

# Exploring the roles of ZmARM gene family in maize development and abiotic stress response

Zhijia Yu[1,2], Xiaopeng Sun[3], Ziqi Chen[2], Qi Wang[2], Chuang Zhang[2], Xiangguo Liu[1,2], Weilin Wu[1] and Yuejia Yin[2]

[1] College of Agriculture, Yanbian University, Jilin, China
[2] Jilin Academy of Agricultural Sciences, Institute of Agricultural Biotechnology, Changchun, China
[3] Huazhong Agricultural University, National Key Laboratory of Crop Genetic Improvement, Wuhan, China

Corresponding authors
Weilin Wu, wlwu@ybu.edu.cn
Yuejia Yin, yyjqishi@163.com

## ABSTRACT

Armadillo (ARM) was a gene family important to plants, with crucial roles in regulating plant growth, development, and stress responses. However, the properties and functions of ARM family members in maize had received limited attention. Therefore, this study employed bioinformatics methods to analyze the structure and evolution of ARM-repeat protein family members in maize. The maize (*Zea mays* L.) genome contains 56 ARM genes distributed over 10 chromosomes, and collinearity analysis indicated 12 pairs of linkage between them. Analysis of the physicochemical properties of ARM proteins showed that most of these proteins were acidic and hydrophilic. According to the number and evolutionary analysis of the ARM genes, the ARM genes in maize can be divided into eight subgroups, and the gene structure and conserved motifs showed similar compositions in each group. The findings shed light on the significant roles of 56 ZmARM domain genes in development and abiotic stress, particularly drought stress. RNA-Seq and qRT-PCR analysis revealed that drought stress exerts an influence on specific members of the ZmARM family, such as *ZmARM4*, *ZmARM12*, *ZmARM34* and *ZmARM36*. The comprehensive profiling of these genes in the whole genome, combined with expression analysis, establishes a foundation for further exploration of plant gene function in the context of abiotic stress and reproductive development.

## INTRODUCTION

Protein repeats are ubiquitous across organisms, serving as small structural units that contribute to the formation of 3D protein structures. Protein tandem repeats are arranged with repetitive sequence units in tandem, which are generated through internal replication and recombination events in genes (*Andrade, Perez-Iratxeta & Ponting, 2001*). The repetition of these small structural units can confer certain advantages to proteins and their respective organisms, and the ARM-repeat protein family represents a highly evolutionarily conserved group of proteins.

The ARM repeats protein is widely distributed among eukaryotes, initially identified in the polar gene fragment of drosophila melanogaster (*Nüsslein-Volhard & Wieschaus, 1980*; *Riggleman, Wieschaus & Schedl, 1989*). Subsequent research revealed the presence of ARM repeats in animal and plant proteins as well. Currently, numerous crystal structures of ARM repeats have been investigated (*Conti et al., 1998*; *Choi & Weis, 2005*; *Valpuesta, 2005*; *Otomo et al., 2005*; *Rose et al., 2005*; *Kidd et al., 2005*; *Tu et al., 2007*; *Liu et al., 2008*; *Zhao et al., 2009*). It has been established that ARM repeat proteins do not necessarily exhibit a high degree of sequence identity. They primarily participate in protein-protein interaction. Proteins containing ARM repeats are known to be involved in many cellular processes, including signal transduction, nuclear transport, cell adhesion, and protein degradation (*Groves & Barford, 1999*; *Stone et al., 2003*; *Coates, 2003*; *Bergler & Hoth, 2011*).

ARM functional domains are often combined with other functional domains to collectively complete their physiological functions. U-box/ARM proteins, the combinations of the U-box and ARM domain, are the largest number of proteins and are the largest family of ARM proteins. U-box protein is considered to be the target protein for degradation (*Azevedo, Santos-Rosa & Shirasu, 2001*). The U-box/ARM Family is unique to higher plants a class of proteins. At present, it is only found in higher plants such as *Arabidopsis thaliana* and rice, because there is no detectable counterpart in other genomes, and means that it may have important functions in higher plants.

The ARM domain, characterized by a superhelical structure composed of several tandem repeat motifs, consists of 42 amino acids in length (*Peifer, Berg & Reynolds, 1994*; *Huber, Nelson & Weis, 1997*), it plays a crucial role in the development of various cellular organisms. In mammals, it is involved in the regulation of gene expression during intercellular adhesion and development (*Logan & Nusse, 2004*; *Nelson & Nusse, 2004*). Studies have revealed that the presence of the ARM repeat domain gives rise to novel functions of ARM proteins in plants. In *Arabidopsis thaliana*, an arm repeat protein involved in abscisic acid signalling (ARIA) (*Kim et al., 2004*). In rice, the Spl11 gene encodes a novel protein with a U-box domain and six armadillo (ARM) repeats. SPL11 has E3 ubiquitin ligase activity *in vitro*, and the U-box domain is essential for E3 ligase activity (*Liu et al., 2012*). There are many other such reports showing that ARM has an important effect on plants.

Maize (*Zea mays* L.), a widely cultivated cereal crop, faces various adversity stresses affecting its yield (*Wang et al., 2019*). Despite extensive research on ARM proteins, some members still have unknown functions. This study aims to investigate the ARM domain protein family in the maize genome, conducting a comprehensive phylogenetic analysis using bioinformatics and publicly available plant databases. Expression patterns of ARM family members during maize development and under abiotic stress were examined. These findings provide insights into the molecular, evolutionary, and functional aspects of ARM proteins in plants.

## MATERIALS AND METHODS

### Plant material and growing conditions

Maize inbred line B73 was used in this study. For stress treatment, B73 seeds were directly sown in the soil at 28 °C in brown pots (10 cm × 10 cm × 9 cm). There are nine inbred plants in both pots, all randomly placed in the growth chamber at 28 °C and light for 16 h/dark for 8 h. When the maize plants reach the three-leaf and one-heart stage, water control is initiated for a portion of the plants, while the remaining portion is watered normally. When the soil moisture content of the water-saving treatment experimental group drops below 20%, the uppermost unfolded leaves, predominantly the third leaf at the V4 stage, are collected for RNA extraction. Most samples obtain two replicates, each containing at least three leaves.

### Identification of ARM family genes in maize

The genome and gene annotation data of maize, Arabidopsis (*Arabidopsis thaliana* (L.) *Heynh*), and rice (*Oryza sativa* L.) were downloaded from the Ensembl database (https://plants.ensembl.org/index.html). The Hidden Markov Model file (Pfam ID: PF00514) of the ZmARM domain genes was found and downloaded in the Pfam protein database (*Mistry et al., 2021*). We chose the version of Ensembl Plants Genes 56 on the website and used the Pfam ID for a similarity search. Finally, we identified 56 ARM proteins by searching maize protein with E < 0.0001 threshold, and screening and identifying the candidate ARM transcription factor domains by database SMART and CDD online.

### Physicochemical properties of ZmARM proteins

We analyzed the physicochemical properties, molecular weight, isoelectric points, amino acid length, aliphatic amino acids, hydrophilicity and hydrophobicity of amino acids were measured using ExPASy's ProtParam. ExPASy ProtParam (*Wilkins et al., 1999*) (http://web.expasy.org/protparm/).

### System evolution analysis method

We studied the evolutionary relationship of the ZmARM domain genes in maize by analyzing ARM repeats of maize, rice, and arabidopsis. In addition, to study the evolutionary relationship of the ARM gene family members of three species, we generated combined phylogenetic trees with the whole protein sequence. Sequence alignment and phylogenetic analyses were carried out by running the software MUSCLE and MEGA11 (*Koichiro, Glen & Sudhir, 2021*), and then the maximum-likelihood (ML) connection algorithm was carried out to construct the trees. The MUSCLE is the default parameter. A total of 100 repeated guided analyses were carried out, and the branch length corresponding to phylogenetic distance was measured by the number of amino acids. The resulting phylogenetic trees were interactively pruned and re-rooted by the online tool iTOL (https://itol.embl.de). These genes are uniformly named in numerical order based on the number of substitutions at each locus, *ZmARM1* to *ZmARM56*.
### Gene structure prediction of ZmARM gene in maize

The coding regions (CDS) and non-coding regions (untranslated regions, UTRs) of ZmARM family members were predicted online with TBtools Visualize Gene Structure.

### Gene structure and motif analysis of the ZmARM family members

We uesd TBtools Introduction-MEME Suite, the protein sequences of ARM family members in maize were analysed (number of conserved structures = 10), Motify prediction analysis was performed based on the amino acid motif, and visual mapping was performed in TBtools Visualize MEME/MAST Patten (*Chen et al., 2020*).

### Chromosome distribution and collinearity between ZmARM family members

We analyzed the orthologous relationship among the maize B73 ZmARM genes using the One Step MCScanX tool in TBtools and analyzed the gene duplication events in the maize B73 (version 5) genome sequences and gene annotations. Furthermore, we generated the collinearity analysis diagram of the ZmARM genes among different species using the advanced tool in TBtools. The Ks (synonymous substitution per) and Ka (nonsynonymous substitution per) parameters of repeat events were calculated using the TBtools calculator (Simple Ka/Ks Calculator), and the Ka/Ks (the ratio of the number of nonsynonymous substitutions per nonsynonymous site (Ka) to the number of synonymous substitutions per synonymous site (Ks)) values were obtained.

### Expression analysis of ZmARM genes family

We downloaded the RNA-seq data of inbred line B73 in different tissues during growth and development in Maize GDB (https://www.maizegdb.org/) (*Walley et al., 2016*) and the data of cold, heat, salt, and ultraviolet stress (*Waters et al., 2017*). We performed a Z-Score normalization of the data. To analyze the expression pattern of ZmARM family members, we drew a heat map with the Heatmap tool in TBtools.

### Expression pattern analysis of ZmARM family members under different drought conditions

The data are come from our laboratory, which are the RNA-seq of B73 in response to drought published by Professor Mingqiu Dai from Huazhong Agricultural University (*Zhang et al., 2019*). Meanwhile, we compared between different RNA-seq data with the same differential expression trend. The expression patterns of the ZmARM gene family under different drought conditions were analyzed, using the Heatmap tool in TBtools to draw a heat map.

### qRT-PCR analysis

The NCBI online primer tool was used to design qRT-PCR-specific primers (https://www.ncbi.nlm.nih.gov/tools/primer-blast/index.cgi?LINK_LOC=BlastHome), and the qRT-PCR primers that were designed were sent to Biotech Biologicals (Wuhan, China) for synthesis (Table S1). ChamQ Universal SYBR qPCR Master Mix (Q711, Vazyme, Nanjing, China) was used for qRT-PCR analysis on a LightCycler machine. Three replicates were

performed for each treatment. The $2^{-\Delta\Delta Ct}$ method was used to calculate relative expression. A t-test was used to compare the differences between the groups.

## RESULTS

### Identification and phylogenetic analysis of ZmARM genes

ZmARM family members were screened from the Ensembl Plant database (https://plants.ensembl.org/index.html), and 56 maize ARM proteins were identified (Table 1). Based on their relationships in the evolutionary tree, 56 maize ARM genes were named *ZmARM1-ZmARM56*. A genetic evolution tree of 56 ZmARM was constructed with software MEGA11.0, and nine subfamilies were classified with 29, 3, 2, 6, 3, 4, 3, 3, 2 members respectively (Fig. 1A).

To study the phylogenetic evolution of ZmARMs, we constructed a phylogenetic tree that included 56 ZmARMs, 54 ARMs (AtARMs), and 40 *Oryza sativa* ARMs (OsARMs) (Fig. 1B). The results showed that these genes were divided into 11 groups, with Group 1 (74 ARMs) and Group 2 (22 ARMs) being the largest groups in terms of the number of ARMs. The smallest subfamilies were Group 9, Group 10, and Group 11, each consisting of two ARMs (Table S1). It is worth noting that the phylogenetic tree had two additional subgroups compared to the genetic evolution tree composed of ZmARM sequences. These results indicate that the ARM family exhibits more diversity in the maize evolutionary tree, suggesting closer kinship in maize during evolution and a certain degree of divergence in the evolution of these three plants.

### Physicochemical properties of ZmARM gene family encodes protein sequences

Based on ExPASy ProtParam analysis, there were differences between these members, but the differences are not significant in the physicochemical properties. The number of amino acids ranged from 117 to 2,144, and the theoretical isoelectric points were in the pH range from 4.51 to 9.51. The predicted isoelectric points indicated that these proteins are mostly acidic, with only 16 proteins encoded by ZmARM being basic. The length of the open reading frames ranged from 351 to 6,432 bases, and the protein molecular weight ranged from 11.99 kDa to 229.97 kDa. The aliphatic index ranged from 83.01 to 115.12, and the protein hydrophilicity ranged from −0.484 to 0.469. Most proteins were hydrophilic (GRAVY < 0) while a few were hydrophobic (GRAVY > 0).

### Chromosomal ocalization of ZmARM genes

We studied the chromosomal localization of the ZmARM gene family (Fig. 2), the analysis showed that they were randomly located on 10 chromosomes, with each gene being located on a unique chromosome (Fig. 2). Every seven ZmARMs were located on chromosomes 2, 3, and 4, which had the highest number of genes, accounting for 37.5% of the total. Additionally, every six ZmARMs were located on chromosomes 1, 6, 7, 8, and 9. Finally, only one ZmARM was located on chromosome 10.

| | Table 1 Gene and protein characteristics of the *ZmARM* members. | | | | | |
|---|---|---|---|---|---|---|
| **Gene name** | **Gene ID** | **Length (aa)** | **MW (Da)** | **pI** | **GRAVY** | **Aliphatic index** |
| ZmARM1 | Zm00001eb012430 | 726 | 78,488.84 | 5.1 | −0.25 | 93.5 |
| ZmARM2 | Zm00001eb397640 | 180 | 18,893.38 | 9.51 | 0.469 | 113.28 |
| ZmARM3 | Zm00001eb029250 | 665 | 72,536.11 | 5.46 | −0.168 | 99.2 |
| ZmARM4 | Zm00001eb136450 | 699 | 75,927.91 | 5.67 | −0.193 | 97.64 |
| ZmARM5 | Zm00001eb379620 | 603 | 65,158.86 | 6.45 | −0.136 | 100.76 |
| ZmARM6 | Zm00001eb190130 | 638 | 68,724.68 | 5.88 | −0.114 | 102.12 |
| ZmARM7 | Zm00001eb241150 | 465 | 50,267.83 | 6.18 | −0.309 | 86.97 |
| ZmARM8 | Zm00001eb033520 | 645 | 71,012.05 | 6.02 | −0.224 | 101.57 |
| ZmARM9 | Zm00001eb228550 | 641 | 71,004.32 | 6.61 | −0.193 | 106.46 |
| ZmARM10 | Zm00001eb178990 | 694 | 73,540.83 | 6.46 | 0.156 | 109.24 |
| ZmARM11 | Zm00001eb311590 | 698 | 73,956.24 | 8.12 | 0.116 | 103.24 |
| ZmARM12 | Zm00001eb239050 | 729 | 78,935.31 | 8.54 | 0.061 | 103.13 |
| ZmARM13 | Zm00001eb204890 | 732 | 79,130.65 | 8.51 | 0.094 | 105.14 |
| ZmARM14 | Zm00001eb290550 | 670 | 70,939.65 | 6.97 | 0.083 | 103.76 |
| ZmARM15 | Zm00001eb149420 | 692 | 73,904.03 | 6.35 | 0.028 | 99.7 |
| ZmARM16 | Zm00001eb368740 | 697 | 74,630.79 | 5.82 | 0.024 | 99.54 |
| ZmARM17 | Zm00001eb326060 | 362 | 37,803.12 | 5.96 | 0.004 | 98.56 |
| ZmARM18 | Zm00001eb420000 | 270 | 28,362.13 | 5.52 | −0.116 | 97.33 |
| ZmARM19 | Zm00001eb085470 | 392 | 41,998.36 | 7.19 | −0.019 | 96.89 |
| ZmARM20 | Zm00001eb107940 | 464 | 47,717.39 | 9.2 | 0.003 | 98 |
| ZmARM21 | Zm00001eb325810 | 465 | 47,687.19 | 6.42 | 0.015 | 97.61 |
| ZmARM22 | Zm00001eb145570 | 800 | 87,576.75 | 5.79 | −0.213 | 97.36 |
| ZmARM23 | Zm00001eb260470 | 872 | 94,364 | 5.83 | −0.174 | 97.11 |
| ZmARM24 | Zm00001eb174100 | 830 | 89,693.5 | 5.74 | −0.213 | 97.22 |
| ZmARM25 | Zm00001eb287530 | 808 | 89,255.07 | 6.4 | −0.17 | 95.94 |
| ZmARM26 | Zm00001eb349910 | 748 | 81,304.71 | 6.03 | −0.137 | 98.82 |
| ZmARM27 | Zm00001eb146820 | 375 | 41,466.05 | 8.21 | 0.065 | 104.77 |
| ZmARM28 | Zm00001eb126230 | 364 | 38,681.54 | 7.65 | −0.07 | 103.87 |
| ZmARM29 | Zm00001eb334600 | 367 | 39,104.1 | 7.02 | −0.078 | 104.6 |
| ZmARM30 | Zm00001eb001150 | 625 | 64,020.84 | 8.9 | 0.135 | 102.7 |
| ZmARM31 | Zm00001eb095530 | 578 | 60,528.1 | 7.64 | 0.32 | 115.12 |
| ZmARM32 | Zm00001eb324200 | 596 | 63,431.79 | 5.22 | −0.066 | 97.32 |
| ZmARM33 | Zm00001eb279480 | 949 | 100,927.16 | 5.36 | 0.151 | 110.47 |
| ZmARM34 | Zm00001eb372570 | 2,144 | 229,971.54 | 5.28 | 0.127 | 109.99 |
| ZmARM35 | Zm00001eb091240 | 2,136 | 229,700.9 | 5.14 | 0.171 | 112.2 |
| ZmARM36 | Zm00001eb166150 | 639 | 68,079.53 | 6.38 | 0.139 | 105.38 |
| ZmARM37 | Zm00001eb115370 | 688 | 73,638.5 | 8.37 | 0.177 | 108.08 |
| ZmARM38 | Zm00001eb336070 | 526 | 56,542.83 | 5.57 | −0.032 | 103.38 |
| ZmARM39 | Zm00001eb331130 | 568 | 61,714.23 | 4.76 | −0.129 | 99.52 |
| ZmARM40 | Zm00001eb127630 | 526 | 57,659.77 | 5.15 | −0.231 | 98.29 |
| ZmARM41 | Zm00001eb333670 | 464 | 50,738.87 | 5.39 | −0.249 | 96.12 |

| Table 1 (continued) | | | | | | |
|---|---|---|---|---|---|---|
| Gene name | Gene ID | Length (aa) | MW (Da) | pI | GRAVY | Aliphatic index |
| ZmARM42 | Zm00001eb282820 | 528 | 58,132.04 | 5.17 | −0.307 | 90.51 |
| ZmARM43 | Zm00001eb346120 | 529 | 58,201.15 | 5.21 | −0.301 | 91.08 |
| ZmARM44 | Zm00001eb224170 | 658 | 69,977.14 | 6.51 | −0.006 | 96.11 |
| ZmARM45 | Zm00001eb398820 | 654 | 70,137.12 | 7.27 | −0.094 | 94.71 |
| ZmARM46 | Zm00001eb103700 | 645 | 70,750.14 | 5.91 | −0.116 | 97.66 |
| ZmARM47 | Zm00001eb190530 | 825 | 89,558.85 | 5.38 | −0.021 | 104.07 |
| ZmARM48 | Zm00001eb141210 | 719 | 76,971.23 | 6.57 | 0.115 | 107.07 |
| ZmARM49 | Zm00001eb270080 | 911 | 99,905.55 | 6.25 | −0.431 | 88.13 |
| ZmARM50 | Zm00001eb377830 | 906 | 99,439.87 | 6.2 | −0.433 | 88.08 |
| ZmARM51 | Zm00001eb004250 | 947 | 105,165.42 | 6.35 | −0.484 | 86.21 |
| ZmARM52 | Zm00001eb188680 | 947 | 105,165.42 | 6.35 | −0.484 | 86.21 |
| ZmARM53 | Zm00001eb311800 | 561 | 57,927.7 | 7.01 | −0.092 | 83.01 |
| ZmARM54 | Zm00001eb099030 | 533 | 54,682.41 | 9.04 | 0.003 | 85.98 |
| ZmARM55 | Zm00001eb023660 | 922 | 98,573.62 | 6.45 | 0.064 | 99.06 |
| ZmARM56 | Zm00001eb390950 | 117 | 11,994.19 | 4.51 | −0.127 | 97.86 |

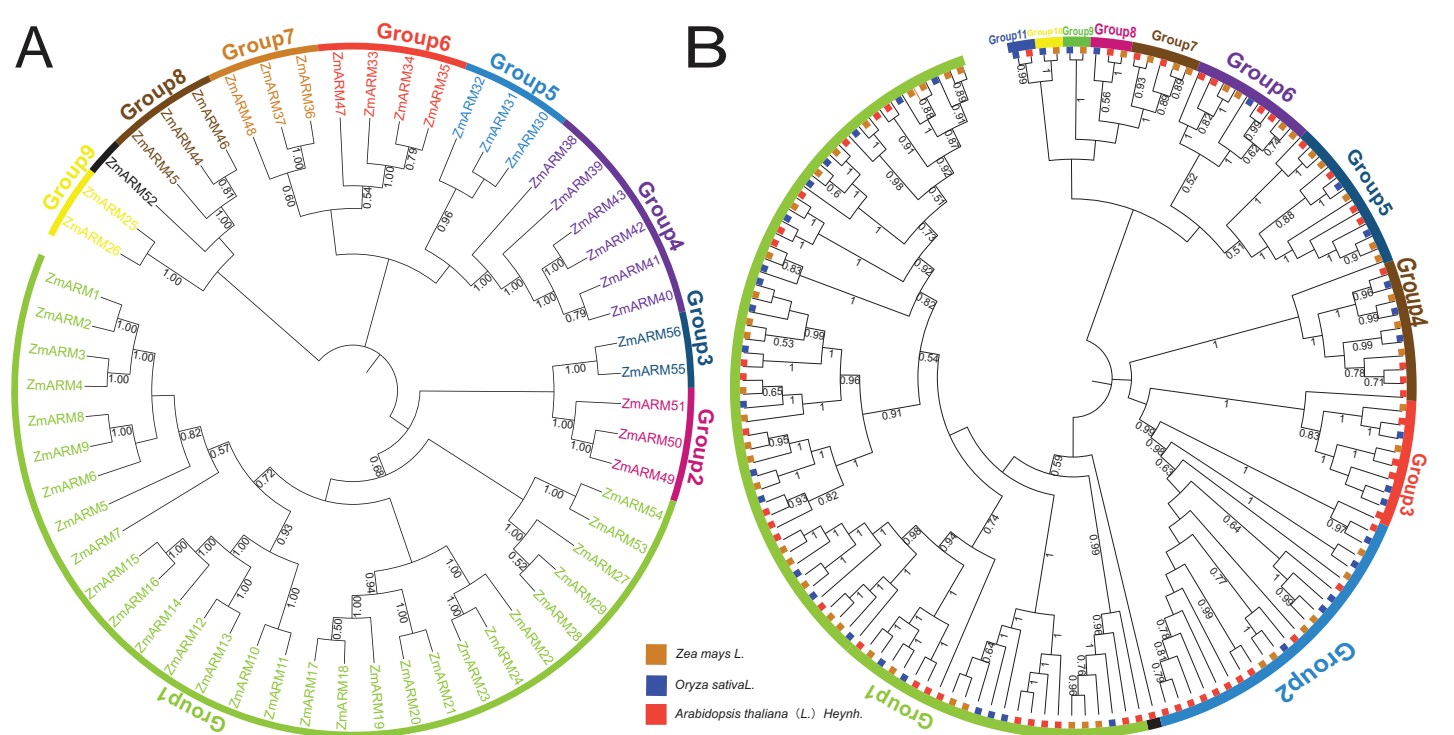

**Figure 1 (A) Phylogenetic analysis of ZmARMs from maize (B) phylogenetic analysis of ZmARMs from maize, Arabidopsis, and rice.** Different subfamilies are represented by different colored arcs. The phylogenetic tree was constructed by running the MEGA X program based on the ML method with 100 bootstrap replications.

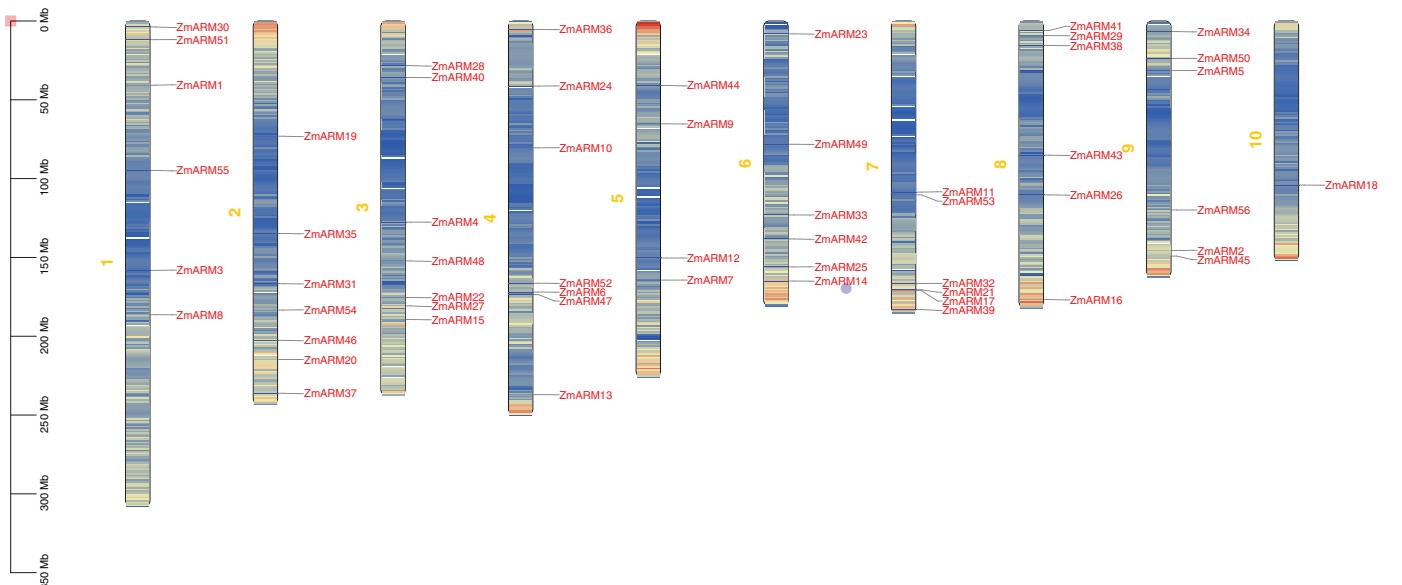

**Figure 2 Chromosomal localization of the ZmARM genes.** Chromosome mapping of 56 ZmARM gene families using TBtools.

## Analysis of gene structure and protein motifs of ZmARM family members

A total of 10 conserved motifs were identified by TBtools to study the homology of ZmARM family members (Fig. 3). We compared the 10 existing motifs on Tomtom (https://meme-suite.org/meme/tools/tomtom) and found 24 known motifs ($q < e^{-3}$), where motif 1, 8, 10 found one known motif, motif 5, 6, 7 found two known motifs, motif 4, 9 found three known motifs, motif 2 found four known motifs, motif 3 found five motifs (Table S2). Sixteen members contained motifs 1, 2, and 8, while 50 members contained motif 8. This indicates that motif 8 is a relatively conserved motif that likely participates in multiple cellular processes.

Structural differences in exon-intron arrangements are an important source of gene family variation and plant diversity. These structural differences lead to variations in gene expression and function. The results show that ZmARM family members are subclassified into eight subgroups on the phylogenetic tree, with some clusters having distinct arrangements, while only a small number of taxa (Fig. 3) share great similarity in exon-intron arrangements. Groups 3 and 4 have the largest number of exons, with more than 10, and the lowest number of ARM family genes (*ZmARM44*, *ZmARM45*, and *ZmARM46*). One member of group 4, *ZmARM33*, has a significantly different exon arrangement compared to other members, as it has only 1 exon. Interestingly, we found that five ARM family gene members (*ZmARM2*, *ZmARM18*, *ZmARM27*, *ZmARM33*, and *ZmARM56*) either lack UTRs or have very short UTR sequences. Most members within the same subgroup exhibit similar motifs and length, indicating functional similarity. The protein sequences within the same subgroup are highly conserved, although there is considerable variation between different groups.
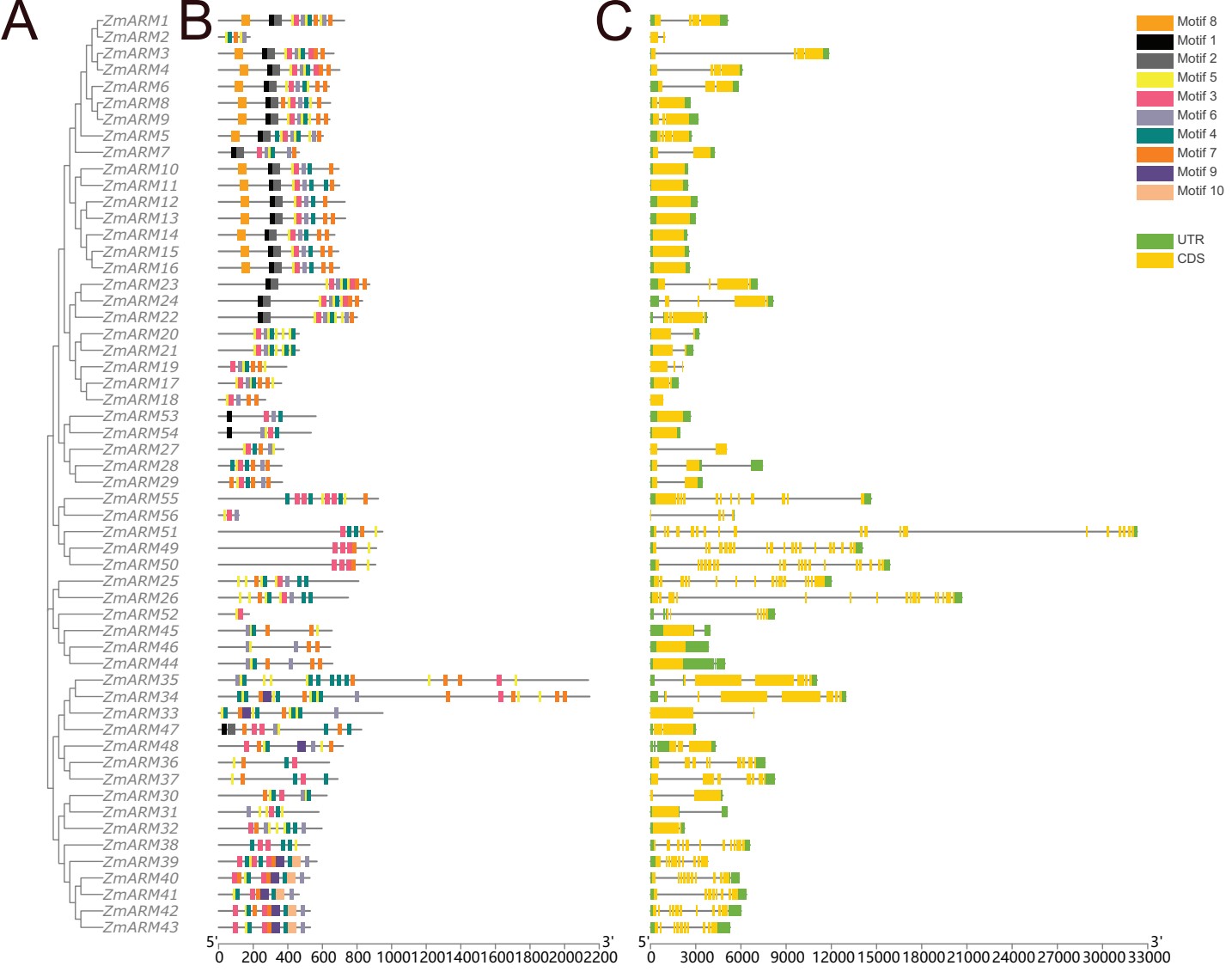

**Figure 3 Gene structure and conserved motifs.** Composition and distribution of conserved motifs in the ZmARM proteins. Conserved motifs are indicated by different numbers and rectangular colors. Exon-intron organization of the ZmARM genes. Exons are shown as yellow rectangles; introns as black lines and UTRS as green rectangles.

## Collinearity among ZmARM family genes

We performed intraspecific MCScanX analysis on maize to gain a clear understanding of the linear relationship between ZmARM family genes within the species. For the analysis of the 56 ZmARM genes using TBtools, we identified 12 pairs of repeated genes, which are as follows:

*ZmARM10-ZmARM11*; *ZmARM13-ZmARM12*; *ZmARM15-ZmARM14*; *ZmARM14-ZmARM16*; *ZmARM15-ZmARM16*; *ZmARM20-ZmARM21*; *ZmARM24-ZmARM23*; *ZmARM28-ZmARM29*; *ZmARM37-ZmARM36*; *ZmARM40-ZmARM41*; *ZmARM41-ZmARM50*; *ZmARM54-ZmARM53* (Fig. 4). The Ks (synonymous substitution per) and Ka (nonsynonymous substitution per) parameters of repeat events were calculated using the

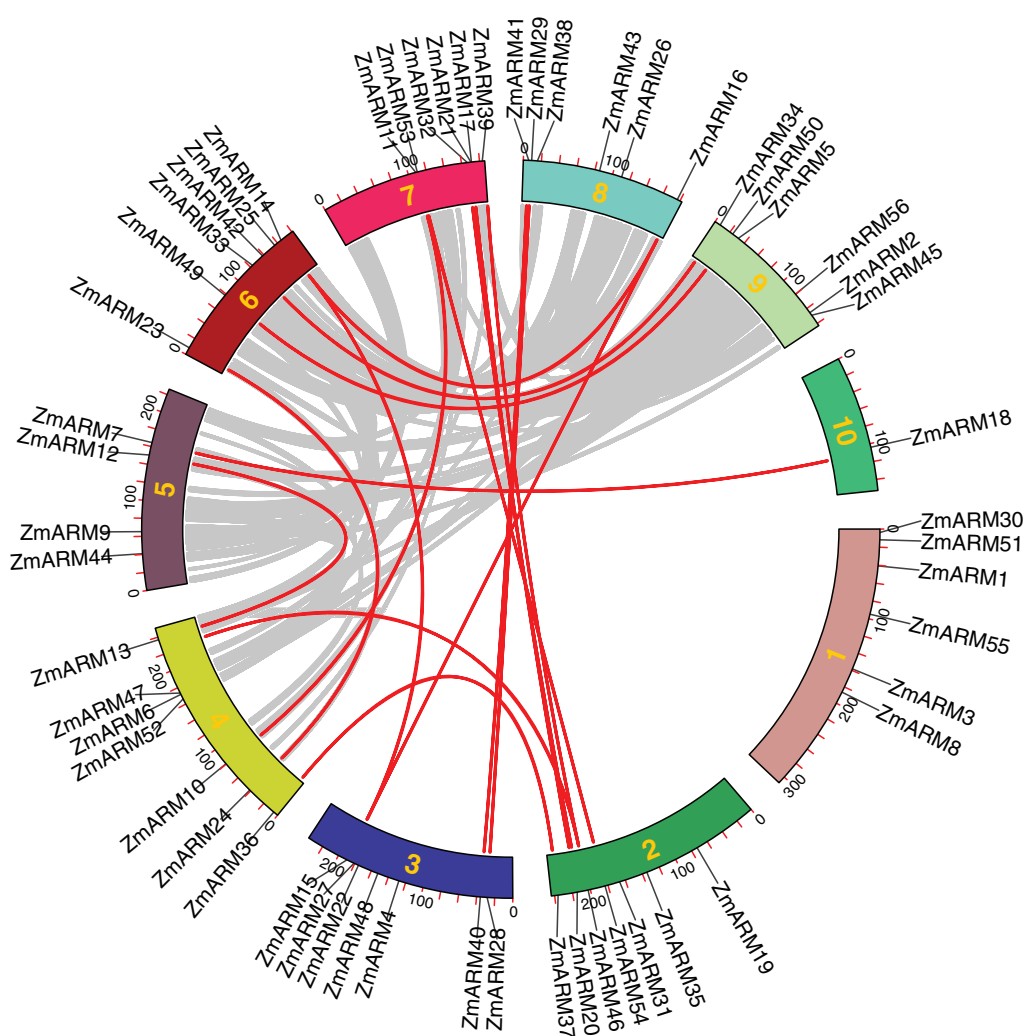

**Figure 4 Collinearity relationships of the ZmARM genes.** Homology analysis of intraspecific genes in ZmARM. Different color rectangles represent chromosomes 1–10, respectively. (The red curve, the homologous gene pairs. Gray lines, collinear gene pairs.)

TBtools calculator (Simple Ka/Ks Calculator). In addition, the Ka/Ks ratio of the 12 ZmARMs tandem repeats were found to be less than 1 (Table S3). Since the Ka/Ks ratio reflects the selection of a gene, these results suggest that the duplicate maize genes underwent purifying selection, which eliminates deleterious mutations in the species.

## Expression analysis of ZmARM genes

To explore the expression of the ZmARM genes in different tissues and developmental stages, we analyzed the expression of the ZmARM gene family. The ZmARMs are tissue-specific expressed genes. Most members of the ZmARM family were highly expressed in immature leaves but had low or no expression in the seeds. Among them, some were highly expressed in roots, such as *ZmARM22* and *ZmARM27*, while others were specifically expressed in pollen, such as *ZmARM4*, *ZmARM6*, *ZmARM11*, *ZmARM24*, *ZmARM29*, *ZmARM46*, and *ZmARM55*. The *ZmARM39* gene did not show expression in

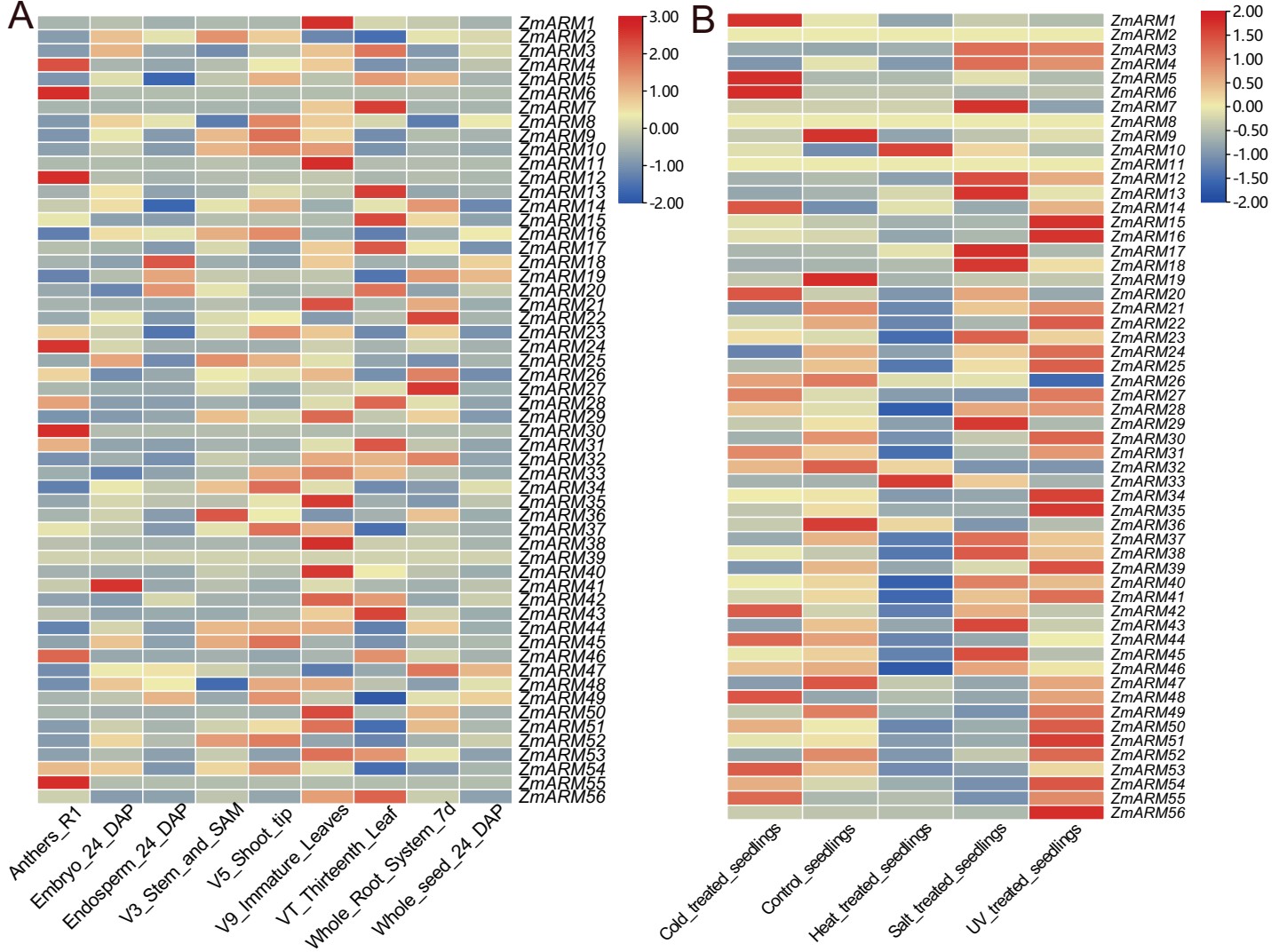

**Figure 5 Expression analysis of the ZmARM genes.** (A) Samples of anthers, embryo, endosperm, stem, SAM, shoot tip, immature leaves, thirteenth leaf, whole root system, and whole seed. (B) Samples under pressure included control seedings, cold-treated seedings, heat-treated seedings, salt-treated seedings, and UV-treated seedings.

the nine stages of maize, indicating that it does not play any roles during the growth or development of maize (Fig. 5A). *ZmARM41* was highly expressed in the embryo, suggesting its potential role in seed development, while *ZmARM18* was highly expressed in the endosperm, suggesting its involvement in seed germination. The presence of *ZmARM36* in the stem tip and stem tip meristem tissue suggests that the gene *ZmARM36* would affect maize development. Expression analysis showed that some ZmARMs were tissue-specific expression genes. For example, *ZmARM4*, *ZmARM6*, *ZmARM12*, *ZmARM30*, and *ZmARM55* were highly expressed in pollen, while *ZmARM41* was expressed specifically in embryos (Fig. 5A).

To explore the expression of ZmARM genes in response to abiotic stresses, we downloaded RNA-seq data (https://maizegd.org) of the inbred line B73 from GDB, and the

data in response to cold, heat, salinity, and UV irradiation, to analyze the expression patterns of ZmARM family members (*Waters et al., 2017*). We analyzed the expression of ZmARM family members under cold, heat, salt, UV and treated seedlings. From the analysis, we found that most of the ZmARM family members showed high expression levels in treated seedlings and UV pressure treatments. Among them, *ZmARM25*, *ZmARM30*, *ZmARM8*, *ZmARM6*, and *ZmARM39* showed high expression under UV stress, with very low or no expression in other abiotic stresses. Interestingly, it was observed that *ZmARM55* exhibited negligible or low expression levels under five abiotic stresses. This suggests that *ZmARM55* may not play a significant role in abiotic stress response. Additionally, a subset of ZmARM family members demonstrated reduced expression levels during seedling treatment. Cold and heat stress exerted some influence on the expression of ZmARM family members, albeit with a relatively small overall impact. In general, the expression patterns of ZmARM genes exhibited specificity in response to abiotic stress conditions. Furthermore, it was observed that 24 ZmARM genes displayed higher expression levels in response to UV treatment-induced stress (Fig. 5B).

## Expression pattern analysis of ZmARM genes under different drought conditions and qRT-PCR validation

Information about gene function can be provided by analyzing the expression levels of the genes. To understand the role of ZmARM family genes under abiotic stress, using drought as an example, we used data from previous studies and genome-wide RNA-seq analysis of B73 self-incompatibility lines to examine the expression profile of ZmARM genes under drought stress treatment (Table S4). Transcriptomic data showed that only 52 of these 56 genes responded to drought stress by changes in expression at three drought stress stages, DT2, DT3, and DT4. The representative meaning of WW (Well Water), DT2 (the soil moistures were 30–35% for DT2), DT3 (20–25% for DT3), and DT4 (10–15% for DT4) can be referred to in the previously published articles (*Zhang et al., 2019*). In the B73 genotype, more than half of the ZmARM genes are in response to drought stress, and the expression of most members tends to be the highest at DT4 drought stage. Heatmap analysis showed that most ZmARM genes showed different up-regulation and expression under drought stress (Fig. 6).

We analyzed the expression of 56 ZmARM genes under drought, and 13 genes significantly differentially expressed at DT4. As shown in the figure, the overall expression level of *ZmARM26, ZmARM30, ZmARM36*, and *ZmARM51* is the highest at DT2, *ZmARM53* is the highest at WW, most ZmARM family gene members in the highest overall expression level at DT2, while at WW, most ZmARMs overall expression level is the lowest, in the 52 genes, only 13 genes showed differential expression between WW and DT4 (Fig. 7). As shown in the 13 genes, eight genes had significantly higher expression levels (*ZmARM4, ZmARM12*, *ZmARM15*, *ZmARM23*, *ZmARM27*, *ZmARM33*, *ZmARM34, ZmARM42*), while only five genes (*ZmARM1, ZmARM29, ZmARM36, ZmARM41*) had significantly decreased expression levels after drought. Judging from the figure, *ZmARM36*, and *ZmARM41* are upregulated at the degree of drought in DT2 and
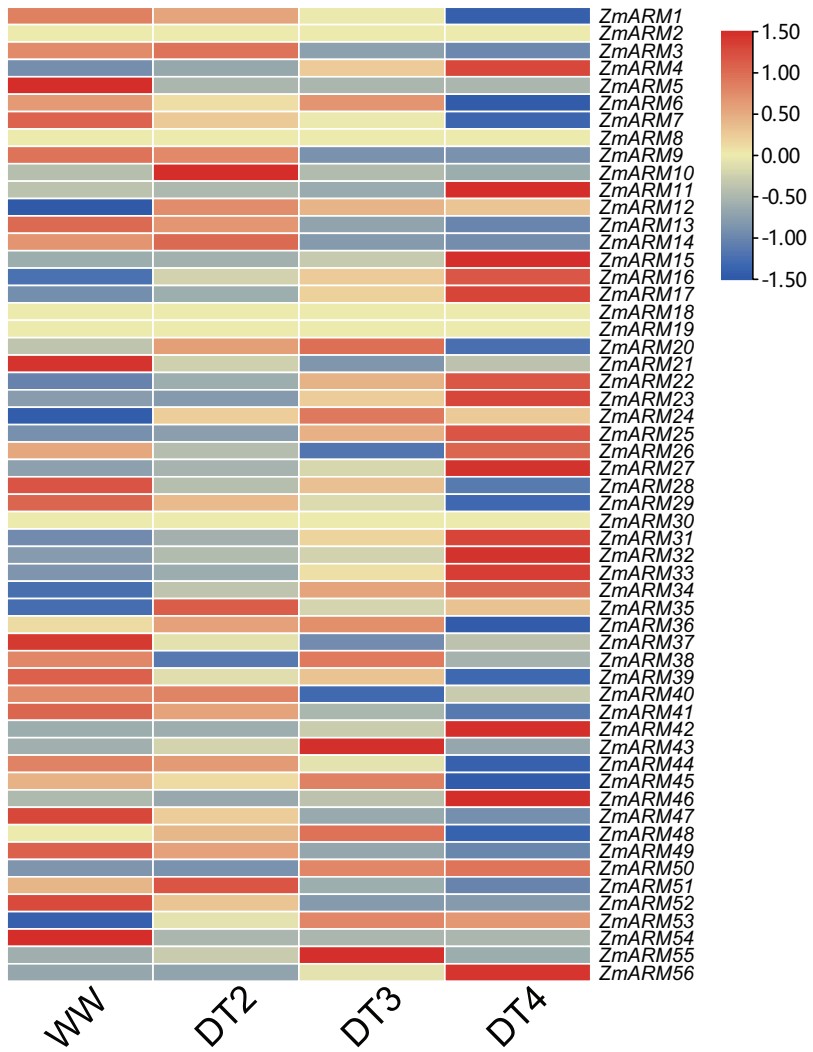

**Figure 6 Expression analysis of ZmARM genes under three different drought degree stresses.**

DT3, but are particularly low at DT4, the other four members had different expression levels at DT3, which are *ZmARM23*, *ZmARM29*, *ZmARM33*, and *ZmARM34* (Fig. 7).

To better understand the expression of these four ZmARM genes under drought stress in maize, we assessed their expression patterns under drought stress using qRT-PCR (Table S5). We found that three of these genes had decreased expression after the drought stress (*ZmARM4*, *ZmARM12*, and *ZmARM34*). However, *ZmARM36* had increased expression after the drought stress.

We performed a comparison of the available drought RNA-seq data with basal leaf meristem drought and maize ovary drought on MaizeGDB (*Kakumanu et al., 2012*). Interestingly, we found that our data were similar in expression changes for eight differential genes for basal leaf meristem drought and ovary drought, while 11 differential ARM genes appeared in the basal leaf meristem drought data and 16 differential ARM genes in the ovaries drought data, we speculate that there is tissue specificity in ARM gene

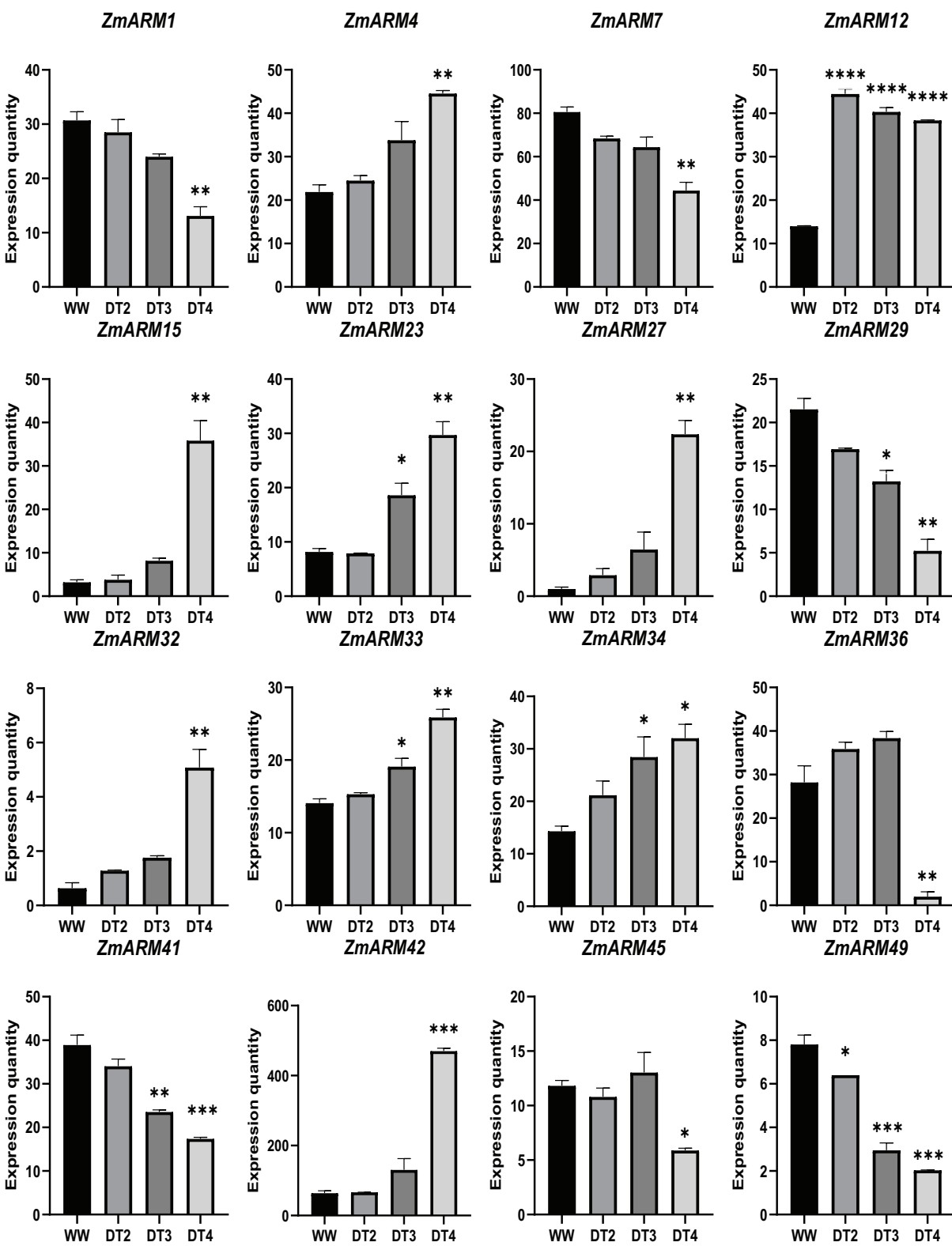

**Figure 7  Differential expression plots of ZmARM genes under three different drought degree stresses.** WW represents CK. ANOVA was used to compare the differences between the groups. *$p < 0.05$, **$p < 0.01$, ***$p < 0.001$, ****$p < 0.0001$.     

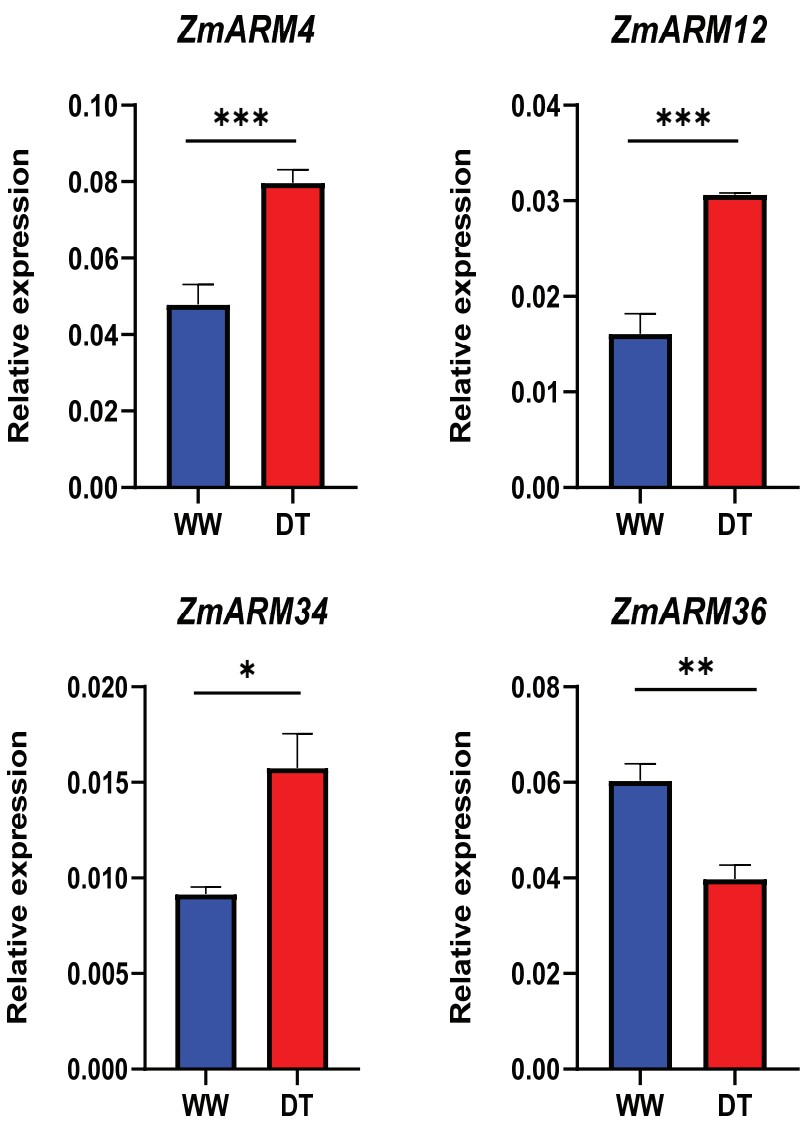

**Figure 8 RT-PCR analysis of ZmARM genes.** Data are the mean ± standard error of three independent replicates. A t-test was used to compare the differences between the groups. $*p < 0.05$, $**p < 0.01$, $***p < 0.001$.

response to drought, indicating that ARM genes have complex roles in plant drought (Table S6).

The results showed that the changes in gene expression detected by both qRT-PCR and RNA-Seq techniques were similar, indicating the reliability of our findings (Fig. 8).

## DISCUSSION

### Organizational forms of ZmARM gene in maize genome

In this study, 56 ARM proteins in the maize genome were identified through a database search. We compared the homologous sequences of ARM proteins in maize, rice, and *Arabidopsis thaliana*, and generated a comprehensive phylogenetic tree. Phylogenetic analysis showed that ARM proteins were distributed in 13 main branches. As expected,

based on overall sequence homology, proteins with similar domain organization tend to cluster. We speculated that *ZmARM36*, and *ZmARM41* may respond to mild drought. In severe drought, it did not work. Interestingly, we found that *ZmARM12* was still significant in all three kinds of drought, the expression of this gene was very induced by drought, so it was speculated that it may be involved in a part of the drought stress pathway. We verified by post-drought RNA-seq and qRT-PCR that the expression of some ZmARM domain genes significantly increased or decreased after drought. These results showed that all 56 selected genes were induced by drought stress, although their expression levels varied after stress.

The results showed that 56 maize ZmARM proteins homologous with *Arabidopsis thaliana* and rice were mainly distributed in groups 1 and 7, which depended on the types and numbers of ARM repeats. DNA replication was one mechanism for improving functional diversity. Diversification of gene function, such as new functionalization or pseudo-functionalization, was often the result of DNA replication events (*Lynch & Conery, 2000*; *Prince & Pickett, 2002*). Previous studies had shown that the function of ARM protein was closely related to its motifs and structures, and the ARM protein mainly participates in various functions of transcriptional regulation and protein interaction, including cell proliferation, hormone regulation, protein transport, and structural scaffold-related functions. One article reported the dual role of U-box/ARM protein 13 in Arabidopsis (PUB 13) in defense and flowering regulation. PUB 13 contains tandem duplication of six (ARM) motifs in its central region and C end. The authors found that PUB 13 encoded U-box/ARM protein repeats with E3 ligase activity, and negatively regulated cell death and $H_2O_2$ accumulation (*Li, Dai & Wang, 2012*). Our study found that the ARM genes in maize were conserved, which were similar to previous studies in Arabidopsis (*Moody et al., 2012*). Overall, most members of ZmARM were affected by drought, so we speculated that ZmARM members will play an important role in regulating plant defense.

## ZmARM genes play an important role in plant development and abiotic stress

The existence of some unique ARM repeats, which played an important role in abiotic stress and plant development, was confirmed by our microarray expression analysis of the ZmARM genes family. Generally, in plant systems, the regulation of protein degradation was related to many pathways, such as light signaling, growth and development, hormone signaling, embryogenesis, leaf senescence, biotic and abiotic stresses (*Yang et al., 2006*; *Drechsel et al., 2011*). The association between U-box proteins and ARM repeats, supported by the fact that several ARM/U-box proteins in Arabidopsis were expressed in different tissues under different growth conditions, suggests that these repeats play an important role in protein degradation and key regulatory pathways (*Samuel et al., 2006*). Interestingly, the expression of some ZmARMs (*ZmARM4*, *ZmARM6*, *ZmARM30*) increased under developmental conditions but decreased significantly under stress conditions. We found specificity from RNA-seq data under different droughts in different tissues and under different drought conditions, which is more speculated that ARM genes

function specifically in response to drought. In addition, it could be speculated that ARM repeats may mediate the interaction with a large number of proteins, thus making the proteasome degradation pathway have substrate diversity. ZmARMs participate in the regulation of plant growth and development.

Plants were affected by multiple abiotic stresses throughout their life cycle, such as drought, extreme temperatures, and high salinity. These factors seriously affect plant growth and development. Thus, plants had evolved complex mechanisms of stress resistance to cope with these adverse growth conditions. Many plant ARM proteins were reported to play crucial roles in responses to multiple environmental stresses. Recently, a new class of ARM repeat proteins had been identified in plants with an E3 ubiquitin ligase motif called the U-box. Previous studies showed that overexpression of *PUB2* and *PUB3* in rice can enhance plant cold tolerance, by maintaining a higher chlorophyll content, ion leakage, and expression level of cold stress-induced marker genes under low temperature, so *PUB2* and *PUB3* also had a positive regulatory role in rice cold response (*Byun et al., 2017*). After low temperature treatment, *CaPUB1* plants had a significantly higher survival rate and chlorophyll content than wild-type plants. Meanwhile, the expression of *DREB1A*, *DREB1B*, *DREB1C*, and Cytochrome P450 genes related to low-temperature stress were also significantly higher than the wild type. This suggests that *CaPUB1* as a positive regulator played an important role in rice response to cold stress (*Min et al., 2016*). Recently, a U-box protein, BrPUBs, associated with temperature stress response, was also identified in rapeseed (*Wang et al., 2015*). The expression of *PUB22* and *PUB23* could be rapidly induced under abiotic stress in Arabidopsis. They interacted with *RPN12a*, and *PUB22* and *PUB23* function in the drought signaling pathway by ubiquitinating *RPN12a* (*Cho et al., 2008*), increased sensitivity to drought stress in transgenic plants overexpressing PUB22 and PUB23. In contrast, loss-of-function *pub22* and *pub23* mutant plants showed significantly enhanced drought tolerance, while the *pub22pub23* double mutant showed increased drought tolerance. These results indicated that *PUB22* and *PUB23* could cooperate to negatively regulate the drought stress response in plants. Our results found that ZmARM families were mostly affected by drought, especially at DT4, where the ZmARM genes had the highest expression. Our expression analysis suggestsed that many ARMs repeat proteins were differentially regulated under stress, possibly indicating their plant-specific functions under stress and developmental conditions, and that their role was conserved in plants.

## CONCLUSIONS

In this study, we identified 56 ZmARM genes across the maize genome. Gene structure and sequence analysis showed that these ZmARM genes, which contain highly conserved ARM structural domains, were unevenly distributed on 10 chromosomes. RNA-seq analysis indicated that these genes may respond to developmental and abiotic stresses in maize. Using drought as an example, post-drought RNA-Seq and qRT-PCR results confirmed that ZmARM genes play an important role in plant drought processes. In conclusion, this study provided a good basis for further studies on the function of ZmARM genes in maize.

## ACKNOWLEDGEMENTS

We thank all the researchers for making the summary data publicly available, and we are grateful for all the investigators and participants who contributed to those studies.

### Funding

This work was supported by funding from the National Natural Science Foundation of China (No. 32171928), the Jilin Provincial Agricultural Innovation Project (No. CXGC2022DX008), the Jilin Provincial Agricultural Innovation Project (No. CXGC2023RCY006), the Agricultural Science and Technology Innovation Program of Jilin Province (No. CXGC2020RCG007), the Industrial Technology Research and development Project of Jilin Province (No. 2022C037-3) and the Technological innovation and breeding application of gene editing of major crops (No. KYJF2023DX006). The funders had no role in study design, data collection and analysis, decision to publish, or preparation of the manuscript.

### Grant Disclosures

The following grant information was disclosed by the authors:
National Natural Science Foundation of China: 32171928.
Jilin Provincial Agricultural Innovation Project: CXGC2022DX008.
Jilin Provincial Agricultural Innovation Project: CXGC2023RCY006.
Agricultural Science and Technology Innovation Program of Jilin Province: CXGC2020RCG007.
Industrial Technology Research and Development Project of Jilin Province: 2022C037-3.
Technological Innovation and Breeding Application of Gene Editing of Major Crops: KYJF2023DX006.

### Competing Interests

The authors declare that they have no competing interests.

### Author Contributions

- Zhijia Yu conceived and designed the experiments, authored or reviewed drafts of the article, and approved the final draft.
- Xiaopeng Sun performed the experiments, analyzed the data, prepared figures and/or tables, and approved the final draft.
- Ziqi Chen analyzed the data, authored or reviewed drafts of the article, and approved the final draft.
- Qi Wang analyzed the data, authored or reviewed drafts of the article, and approved the final draft.
- Chuang Zhang analyzed the data, authored or reviewed drafts of the article, and approved the final draft.

- Xiangguo Liu conceived and designed the experiments, authored or reviewed drafts of the article, and approved the final draft.
- Weilin Wu conceived and designed the experiments, authored or reviewed drafts of the article, and approved the final draft.
- Yuejia Yin conceived and designed the experiments, performed the experiments, analyzed the data, prepared figures and/or tables, and approved the final draft.

## DNA Deposition

The following information was supplied regarding the deposition of DNA sequences:

All the gene numbers used in this study are available in the Ensembl Plants database: Zm00001eb190130, Zm00001eb149420, Zm00001eb145570, Zm00001eb324200, Zm00001eb279480, Zm00001eb282820, Zm00001eb398820, Zm00001eb141210. https://plants.ensembl.org/index.html.

The RNA-seq data is available at NCBI GEO: GSE124340 (expression pattern analysis of ZmARM genes under different drought conditions), PRJNA171684 (different tissues during growth and development in Maize GDB), PRJNA244661 (the expression of ZmARM genes in response to abiotic stresses) and GSE40070 (performed a comparison of the available drought RNA-seq data with basal leaf meristem drought and maize ovary drought on MaizeGDB).

## Data Availability

The raw data is available in the Supplemental Files.

## Supplemental Information

Supplemental information for this article can be found online at http://dx.doi.org/10.7717/peerj.16254#supplemental-information.

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
