# Peer review of "Exploring the roles of ZmARM gene family in maize development and abiotic stress response"

_PeerJ, doi:10.7717/peerj.16254_

## Round 0.1 · original submission · Major Revisions

Dear authors, I thank you for your contribution. I would appreciate it if you would revise the manuscript after reviewing the feedback from the reviewers.

Reviewer 1 ·

Basic reporting

The manuscript "Exploring the roles of ZmARM gene family in maize development and abiotic stress response" conducts an exhaustive study on ZmARM genes in maize, deploying a range of bioinformatics tools, RNA-seq data examination, and qRT-PCR for verification. The authors adeptly demonstrate their proficiency in the use of various bioinformatics tools. Broadly, the authors effectively communicate their findings and substantiate their claims with pertinent data. However, the manuscript incorporates an extensive suite of information from various bioinformatics analyses and results, which may challenge the readers in comprehending the primary objectives of the study. A more refined approach, focusing narrowly on a few pivotal results that closely align with the overall aim of this manuscript, i.e., "examining the properties and functions of ARM genes/proteins in maize", could significantly bolster the manuscript's impact. Additionally, several minor issues relating to readability and text clarity have been noted.
Major Concern:
• The overarching objective of the manuscript could benefit from clearer articulation. The current breadth of bioinformatics analyses incorporated in the study has the potential to obscure the central aim. As it stands, discerning whether the research primarily concentrates on the exploration of ARM genes in maize or embarks on a broader inquiry into ARM evolution across different crops is challenging. As stated in Lines 78 to 80, the manuscript's primary objective—to investigate the ARM domain protein family in maize through a comprehensive phylogenetic analysis—necessitates enhanced focus and clarity. At present, the wide array of bioinformatics analyses included, some of which appear peripheral to the main objective, may lead to reader confusion and diminish the resonance of the principal findings. To improve the manuscript's clarity and the impact of the research, a more streamlined approach, focusing predominantly on analyses directly pertinent to the study of the ARM protein family in maize, is advisable.
Minor Concerns:
1. Line 34: “…comprehensive identification…” may be rephrased as “comprehensive profiling”. The authors clearly performed more analysis than “identification” of ZmARM genes.
2. Line 124: revise “non-coding regions (UTR)” as “non-coding regions (untranslated regions, UTRs)”. If the term “UTRs” is defined in line 124, in Figure 3 caption, “untranslated regions (UTR)” should be “UTRs”.
3. Missing ZmARM56 row on Table 1
4. Line 166: acronym AtARMs and OsARMs need definitions within the paragraph.
5. Delete orphan figure and table captions: Line 175 to 178, Line 191, Line 200 to 201, Line 223 to 227, Line 245 to 248, Line 284 to 288, Line 304 to 305, Line 319 to 321, and Line 330 to 331.
6. Line 180 and Line 182: “physiochemical” appears to be a typo of “physicochemical”.
7. Line 184: “PH” should be “pH”
8. Line 204 and Figure 3: “10 conserved motifs” - What exactly are these motifs? I would suggest the authors put them on Figure 3 or provide a table of these motifs in the supplementary and add a reference here so that interested readers can explore them further.
9. Line 206: “…that motif 4 is a relatively…” - How many members contained motif 4; or did the authors mean motif 8 here?
10. Line 239: the authors need to unify the term “KS/KA”. It seems instances of “Ka/Ks” are used mor frequently, the authors may want to replace “KS/KA” with “Ka/Ks”.
11. Line 251: Add reference to Figure 5 here.
12. Line 252: It is not clear what “evolutionary results” do the authors refer to here.
13. Line 282 to 283 “…it was observed that ZmARM genes displayed higher expression…” If the authors would claim all ZmARM genes displayed higher expression, this statement seems to contradict with previous one in Line 276, where authors stated that at ZmARM55 exhibited negligible or low expression levels. – so how many ZmARM genes displayed higher expression here?
14. Line 298: “of ’WW, DT2, DT3, and DT4’” - For better readability, the authors should consider to add some details on what these drought conditions are, even if this experiment refers to previously published articles.
15. Line 336 to 337: With the presumption of defined study scope as “ZmARM genes in maize”, the purpose of including ARM proteins in rice and Arabidopsis in this study is unclear. Why is sequence homology analysis needed? Is it because ARM proteins in rice and Arabidopsis thaliana are better studied? The authors may want to further discuss this topic in the discussion section.
16. Figure 4 caption: “intr-specific” appears to be a typo of “intraspecific”

Experimental design

The experimental design, which involved the utilization of RNA-seq data, bioinformatics tools, and qRT-PCR validation, is solid and well-executed. The authors have shown a robust understanding and execution of the methodologies involved. However, it appears that some aspects of the experimental design could be expanded upon to provide a more comprehensive understanding of the role of ZmARM genes in maize.
Minor Concern:
• More clarity could be provided regarding the selection of tissues and developmental stages for RNA-seq analysis. How were these particular tissues and stages chosen, and what is the rationale behind this selection?

Validity of the findings

The findings in the manuscript are well presented, and the use of RNA-seq data coupled with qRT-PCR validation strengthens their validity. However, to establish more solid findings and contribute to the comprehensive understanding of ZmARM genes in maize, further analysis could be considered.
Major Concerns:
1. Given the extensive use of public data from plant databases in the bioinformatics analysis, a comparison of the authors’ experimental findings (Line 145) with data available in the MaizeGDB (Line 138 to 139) would strengthen the validity and potential applications of this study. Such comparison would not only serve as a form of cross-validation but also provide a comprehensive context for the authors’ findings, making this study more robust and reliable. Please consider incorporating this comparison into the results and discussion sections.
2. It is admirable that the authors have taken the effort to include a comprehensive list of sequences in the supplementary file titled "The_protein_sequences_used_in_the_phylogenetic_analysis.txt". However, an apparent discrepancy arises upon observing that the list contains 148 ZmARM sequences, which is nearly double the 56 ZmARM genes that the authors have been referencing. Could this difference be attributed to the presence of sequence variants within the ZmARM gene family or to a potential redundancy within the sequence database? It's widely acknowledged that genomes of many organisms exhibit genetic redundancy, wherein multiple genes perform similar roles. This feature, if not carefully accounted for, could be erroneously interpreted as collinearity and may confound the interpretation of the collinearity analysis results, making the deciphering of gene-gene correlations more complex. The authors' clarification on this matter would be greatly appreciated.

Additional comments

This manuscript provides an extensive examination of the ZmARM gene family in maize, presenting a wealth of findings derived from complex bioinformatics analyses. However, the presentation and clarity of these findings could be improved to ensure their full appreciation and understanding by a diverse readership.
In several instances, technical terms and abbreviations were employed without adequate definition or explanation. This makes it difficult for readers who are not intimately familiar with this field to follow the presented arguments and data. To enhance accessibility and readability, all technical terms and abbreviations should be clearly defined upon first use.
Furthermore, the manuscript suffers from an excess of information from various bioinformatics analyses, some of which may not directly contribute to the main objective of the study. Streamlining the content to focus more specifically on the roles of ZmARM genes in maize would be beneficial.
While the manuscript has the potential to make a valuable contribution to the field, the issues outlined above necessitate significant revisions to improve clarity, readability, and focus. Once these revisions are addressed, the manuscript could offer more significant insights into the roles of ZmARM genes in maize.

Reviewer 2 ·

Basic reporting

In the manuscript entitled “Exploring the roles of ZmARM gene family in maize development and abiotic stress response”, the authors performed bioinformatic analyses and expression analyses of the ARM gene family in maize. However, there are some major concerns, including oversimplified introduction, contradictory statements, improperly applied statistical tests, outdated multiple sequence alignment and phylogenetic analysis, and incomplete method section, which significantly weaken the reliability and validity of the findings.

Here are some comments to help to improve the quality of this manuscript.
1. The introduction is too shallow and oversimplified. The authors need to thoroughly introduce the previous research of the structure, function, and expression pattern of ARM proteins in various organisms.
2. The following two statements are contradictory against each other:
Line 20–21, “Armadillo (ARM) was a gene family UNIQUE to plants, with crucial roles in regulating plant growth, development, and stress responses.”
Line 46–49, “The ARM protein is WIDELY DISTRIBUTED AMONG EUKARYOTES, initially identified in the polar gene fragment of drosophila melanogaster (Nüsslein-Volhard & Wieschaus, 1980; Riggleman, Wieschaus & Schedl, 1989). Subsequent research revealed the presence of ARM repeats in animal and plant proteins as well.”
The authors need to make sure that ALL the statements in the Introduction accurately describe the current knowledge.
3. The writing of this manuscript would need substantial improvement. There are many sentences with ambiguity, for example, line 69–71, “Within this domain, a subset of beta-catenin and nuclear transporter proteins are conserved across eukaryotes (Mudgil et al., 2004).” This sentence is difficult to understand. Proteins cannot be a subset of a domain.
Another example is line 71–74, “It serves as a homolog to the ARM protein and plays a crucial role in the development of various cellular organisms. In mammals, it is involved in the regulation of gene expression during intercellular adhesion and development (Logan & Nusse, 2004; Nelson & Nusse, 2004).” What does the “it” refer to? The writing needs to be concise.
The above two sentences are just examples to improve. There are many other sentences need improvement to make the point clear and concise. The reviewer wants to make it clear that the authors need to substantially improve the scientific language throughout the whole manuscript, not just those examples listed by the reviewer.
4. Some sentences are not grammatically correct, for example, line 127–130, “Using TBtools Introduction-MEME Suite analyzed the protein sequences of ARM family members in maize, (the number of conserved structures = 10), carried out Motify prediction analysis based on the amino acid motif, and made visual mapping in TBtools Visualize MEME/MAST Patten”. Also line 138–140, “Downloading the RNA-seq data of inbred line B73 in different tissues during growth and development in Maize GDB (https://maizegd.org) including (Walley et al., 2016) the data of cold, heat, salt, and ultraviolet stress (Waters et al., 2017).” There are many sentences like this throughout the manuscript. The sentences need a subject and a verb. Again, the reviewer wants to make it clear that the authors need to correct the grammar throughout the whole manuscript, not just those examples listed by the reviewer.

5. Line 102, “corn” should be “maize”. It is not professional to use “corn” and “maize” interchangeably in academic writing.
6. Line 99 and 111, “domain” should not be underscored.
7. The subheadings were not formatted properly. For example, line 143–144, “Expression Pattern Analysis of ZmARM family members under Different Drought Conditions”, and many other subheadings. The words were randomly capitalized.
8. Line 250–251, “To explore the expression of the ZmARM genes in different tissues and developmental stages, we analyzed the expression of the ZmARM gene family.” From what data? With which method?
9. The reference list is problematic, which was not arranged in any reasonable order. And some journal names were not accurate. For example, “The Plant Journal” is not called “The Plant Journal: For Cell and Molecular Biology”. The reviewer understands that PeerJ has a special policy allowing the authors to choose whatever citation format they prefer at first submission, but the authors should at least use a professionally arranged format from a citation management software, as a bare minimum.
10. The authors stated they identified 56 ARM genes, but there are 55 ARM genes in Table 1, which is contradictory against each other.
11. The authors need to have a unit for the length in Table 1, “aa”, so that it is clear that the length refers to protein length.
12. The resolution and font size of the figures need to be increased.
13. For Figure 5 and Figure 6, the heatmaps, the values should be normalized z-scores of the log2 expression value, not just the log2 expression value.
14. For Figure 5, the authors need to list each of the data accession codes of the RNA-seq data used in the analysis, not just “Downloading the RNA-seq data of inbred line B73 in different tissues during growth and development in Maize GDB (https://maizegd.org) including (Walley et al., 2016) the data of cold, heat, salt, and ultraviolet stress (Waters et al., 2017).”
15. A different RNA-seq dataset was used in Figure 6. But this RNA-seq dataset was not deposited in any public accessible databases, which does not fulfill the data availability requirements of PeerJ.

Experimental design

1. As described in the Methods, “Sequence alignment and phylogenetic analyses were carried out by running the software ClustalW and MEGA11. (Koichiro, Glen & Sudhir, 2021) , and then the neighborhood connection algorithm was carried out to construct the trees.”
First, there is no such algorithm termed “neighborhood connection algorithm”. It is termed “neighbor-joining algorithm”. Please use widely accepted scientific terms in the manuscript, instead of using machine-translated languages.
Second, with the remarkable advances of multiple sequence alignment (MSA) algorithms and phylogenetic algorithms in the past few decades, it is not acceptable to use ClustalW for MSA or NJ for phylogenetic tree anymore, because they do not produce accurate results, as compared with many other recently developed algorithms. Although many publications still use these algorithms, unfortunately, that does mean it is the correct way to perform the analyses. Here is a benchmarking study for MSA algorithms, and obviously ClustalW was ranked among the poorest performing algorithms: Pais, F. S. M., Ruy, P. D. C., Oliveira, G., & Coimbra, R. S. (2014). Assessing the efficiency of multiple sequence alignment programs. Algorithms for molecular biology, 9, 1-8. The reviewer suggests that the authors use an MSA algorithm with better performance.
NJ method is a distance-based clustering method and is acceptable to produce a quick analysis, but it does not generate publication quality trees. The reviewer suggests the authors to construct the phylogenetic tree using maximum-likelihood (ML) based algorithms. The authors can refer to this paper for more information about the comparison of different phylogenetic analysis algorithms: Zhou, et al. "Evaluating fast maximum likelihood-based phylogenetic programs using empirical phylogenomic data sets." Molecular biology and evolution 35.2 (2018): 486-503. (The Zhou et al., 2018 paper is recommended by the reviewer to demonstrate why ML method should be used, but this paper itself does not need to be cited in this manuscript.) Among the reliable phylogenetic tree construction algorithms, IQ-TREE is one of best algorithms to use, and it has an easy-to-use web-based interface. In short, since the phylogenetic analysis is critical for this study, the authors should use a ML-based program like IQ-TREE to produce a more reliable tree with publication quality and describe the parameters in detail.
2. The statistical analysis was not properly applied. For Figure 7, ANOVA should be used, instead of t-test, as there are several groups.
3. For Figure 8, the authors need to describe the statistical test.
The Method section is incomplete, and does not have sufficient details, especially for the expression analysis of ZmARM genes. And there are many analyses not mentioned at all in the Methods, for example, the Ka/Ks ratio calculation. For all the results presented in the manuscript, the authors need to describe the methods in detail. Otherwise, the validity of the results cannot be assessed.

Validity of the findings

The reviewer has serious concerns regarding the validity of the findings, due to the oversimplified introduction, contradictory statements, improperly applied statistical tests, outdated multiple sequence alignment and phylogenetic analysis, and incomplete method section. If the authors are willing to address all the concerns, the reviewer would be willing to reassess the validity of the revised manuscript.

---

## Round 0.2 · Minor Revisions

The authors have revised the manuscript extensively and improved it to an acceptable level. Still, there is room for improvement in the manuscript as suggested by the reviewers. Please revise the manuscript on the suggested line.

Reviewer 1 ·

Basic reporting

No more comments.

Experimental design

No more comments.

Validity of the findings

No more comments.

Additional comments

I have reviewed the revised version of the manuscript titled "Exploring the roles of ZmARM gene family in maize development and abiotic stress response" submitted by Zhijia Yu, et. al for PeerJ. The authors have effectively addressed all the concerns raised during the initial review.

The revisions have enhanced the quality of the manuscript, and I believe it now meets the standards required for publication in PeerJ. Therefore, I recommend its acceptance for publication.

Reviewer 2 ·

Basic reporting

In the revised manuscript entitled “Exploring the roles of ZmARM gene family in maize development and abiotic stress response”, the authors addressed some of the reviewers’ concerns to improve the quality of the manuscript. However, there are some concerns not being addressed in the revised version.
1. In the previous review, the reviewer requested the authors to finish their Methods section by describing the methods for all of the analysis involved in this study. Please find the request in the previous review: “The Method section is incomplete, and does not have sufficient details, especially for the expression analysis of ZmARM genes. And there are many analyses not mentioned at all in the Methods, for example, the Ka/Ks ratio calculation. For all the results presented in the manuscript, the authors need to describe the methods in detail. Otherwise, the validity of the results cannot be assessed.” It is a requirement to have a complete Methods section. The authors need to describe the detailed methods for ALL of the analysis mentioned in the manuscript, for example the Ka/Ks ratio calculation.
2. As the reviewer mentioned in the previous review, “The reference list is problematic, which was not arranged in any reasonable order. The reviewer understands that PeerJ has a special policy allowing the authors to choose whatever citation format they prefer at first submission, but the authors should at least use a professionally arranged format from a citation management software, as a bare minimum.” This issue was not fixed. If the authors are arranging the references based on the sequence of occurrences, then these should be listed with numbers. Otherwise, the reference list should be arranged based on the name of the first author.
3. The reviewer requested the authors to include accession numbers for each of the datasets used in the analysis. The authors added the following statement, “All the RNA-seq data can be publicly reached in NCBI GEO with accession number: GSE124340, PRJNA171684, PRJNA244661 and GSE40070.” This is too simplified to cite datasets from other publications. The authors used multiple datasets for multiple analyses in this study. These should be listed in more detail to include which dataset was used for which analysis. Again, this comes back to the problem of being too simplified in describing the methods and datasets.
4. The reviewer requested the authors to proofread the manuscript. The authors fixed many of the minor issues, which is great. But there are still some minor issues like spelling errors or improper italics:
Line 171, “Arabidopsis thaliana” should be italicized while “rice” should not. And the authors should use Oryza sativa instead of rice.
Line 193, “Chromosomal iocalization”;
Line 201, “membersb”.
All the gene names should be italicized, while protein names should not.
Please proofread the entire manuscript more carefully to fix any remaining typographical and grammatical issues.

Experimental design

See above. The concerns are still about the over simplified method section.

Validity of the findings

See above.

---

## Round 0.3 · accepted · Accept

Revised version of the manuscript is acceptable.

For instance: "Armadillo (ARM) is a gene family...